# Improving the Process of Early-Warning Detection and Identifying the Most Affected Markets: Evidence from Subprime Mortgage Crisis and COVID-19 Outbreak—Application to American Stock Markets

**DOI:** 10.3390/e25010070

**Published:** 2022-12-30

**Authors:** Heba Elsegai

**Affiliations:** Department of Applied Statistics, Faculty of Commerce, Mansoura University, Mansoura City 35516, Egypt; dr.heba.elsegai@mans.edu.eg

**Keywords:** complex systems, network analysis, financial markets, expectation-maximization algorithm and kalman filter, Granger-causality analysis

## Abstract

Stock-market-crash predictability is of particular interest in the field of financial time-series analysis. Famous examples of major stock-market crashes are the real-estate bubble in 2008 and COVID-19 in 2020. Several studies have studied the prediction process without taking into consideration which markets might be falling into a crisis. To this end, a combination analysis is utilized in this manuscript. Firstly, the auto-regressive estimation (ARE) algorithm is successfully applied to electroencephalography (EEG) brain data for detecting diseases. The ARE algorithm is employed based on state-space modelling, which applies the expectation-maximization algorithm and Kalman filter. This manuscript introduces its application, for the first time, to stock-market data. For this purpose, a time-evolving interaction surface is constructed to observe the change in the surface topology. This enables tracking of the stock market’s behavior over time and differentiates between different states. This provides a deep understanding of the underlying system behavior before, during, and after a crisis. Different patterns of the stock-market movements are recognized, providing novel information regarding detecting an early-warning sign. Secondly, a Granger-causality time-domain technique, called directed partial correlation, is employed to infer the underlying interconnectivity structure among markets. This information is crucial for investors and market players, enabling them to differentiate between those markets which will fall in a catastrophic loss, and those which will not. Consequently, they can make successful decisions towards selecting less risky portfolios, which guarantees lower losses. The results showed the effectiveness of the use of this methodology in the framework of the process of early-warning detection.

## 1. Introduction

A stock price crash is a phenomenon which occurs in the stock market in which a stock index or individual stock price falls sharply within a short time period, [1]. Therefore, predicting crashes in the stock-market system has been the focus of numerous studies [1,2,3,4,5,6,7,8]. It is well-known that the existing literature on stock-market crashes is extensive. Numerous studies have focused on the detection of early-warning signals of market distress from option contracts [9,10,11,12,13]. In addition, some studies have focused on predicting stock crashes at the firm level [4,7], while others have studied the construction of generic indicators to capture critical transitions in the system, as in ecology and climate science [2,3,6,8]; however, the milestone research was conducted by Scheffer et al. [14]. Some studies have employed the concept of capturing critical transitions in complex systems for the purpose of constructing correlation indicators [2,15]. Numerous studies have applied multi-fractal methods to financial time-series data [16,17,18,19]. In addition, there is interest in the use of artificial intelligence for the detection of crisis scenarios [20,21,22,23]. Social-media data may also be used for the early detection of emergency events [24,25].

Almost all the work carried out has only focused on predicting whether there is potential, and there has been little research on the indication of which specific markets are affected most by a crisis. Therefore, the current work expands upon the previous literature in the following important aspects. We start by constructing the interaction parameter space for each period under study separately, then analyze the change of topology in each of them. This is done for the purpose of observing the co-movement motion behavior of the stock markets, for the comparison of different periods. Then, critical transitions are captured, along with an analysis of the relationships between the markets where strong interactions are detected. These analyzed interactions correspond to the high-density regions captured in the reconstructed spaces.

To this aim, we utilize the auto-regressive estimation (ARE) method introduced by Schelter (2014) [26,27], which is a mathematical framework to uncover the time-dependent interaction structure, from measured data, of arbitrary non-stationary stochastic models. The results are presented as spaces, constructed in terms of the high density detected among different processes, which refer to strong interactions. The aim of this approach is to estimate the model parameters without using the windowing approach, avoiding the potential problems associated with the latter. This provides a deeper understanding of the dynamical pattern and the interactions inherited in such systems [26,27]. The ARE algorithm is based on applying the expectation_maximization (EM) algorithm [28] based on the Kalman filter (KF) [29] for autoregressive parameter estimation. This algorithm provides an iterative maximum likelihood estimator [30] for SSM model parameter estimation [26,31]. Explicitly, this maximum likelihood approach accounts for observational noise using SSM which, in turn, provides unbiased estimators [28].

In addition, a Kalman filter is utilized in order to obtain estimates of the hidden states which, in turn, are used to improve the process parameters estimates. The use of the EM algorithm is robust in this setting, which refers to its more stable convergence, ensuring the stability of the model [26]. The usefulness of using SSM together with the EM algorithm has been demonstrated [32,33]. This algorithm has been successfully applied to electroencephalography (EEG) [34] and magnetoencephalography (MEG) data [35]. The main idea behind this algorithm is that it depends only on the estimated autoregressive coefficients using EM and KF to reconstruct the underlying interaction parameter space. This reconstruction provides a clear idea of the behavior of market motions. This part of the analysis accounts for the time-varying pattern of the transmission of stock-price shocks.

On the other hand, several statistical analysis techniques have been developed to detect relationships in multi-variate systems. Examples of such techniques are based on mutual information [36,37,38,39,40], autoregressive processes [41,42,43], coherence [44,45,46], and recurrence in the state space [47,48,49]. A typical assumption to be made when estimating the causality structure from measured data is stationarity; however, the underlying stock-market behavior is, in fact, governed by time-dependent dynamics, such that non-stationarity is present. This violates the assumptions underlying the standard techniques, which are usually based on the concept of Granger causality, originating from econometrics [50,51,52,53,54]. The most well-known frequency- and time-domain techniques based on this concept are renormalized partial directed coherence (rPDC) [55] and directed partial correlation (DPC) [56].

A widely used approach to remove spurious trends related to the random walk is based on taking the logarithm of stock-market returns, instead of working with prices (i.e. raw data) [57,58,59,60,61,62]. In addition, the moving-window approach allows for time-dependent parameters to be estimated [63]; however, the main problem associated with this approach is the appropriate choice of the time-window length, as well as whether there is an overlap between the windows [61,62,63]. In general, if the window size is not chosen properly, then this approach may lead to an under- or an over-estimation of the inferred interactions [63]. In this case, therefore, other approaches should be used.

In particular, financial markets are well-known for being characterized by non-Gaussian distributions, as their fluctuations typically present tails in the case of short returns. However, the long-time returns follow Gaussian distributions. They are often obscured by a large amount of observational noise, which is assumed to be Gaussian noise. This observational noise is not a part of the dynamics of the process; on the contrary, dynamic noise is added to the dynamics of the process. Based on this, both types of noise can be taken into consideration under stochastic models using SSM. The SSM consists of two equations: one that describes the dynamics of the process, as well as an observation equation that models the observation function and observational noise. Such models require accurate parameter estimates; however, existing naïve parameter estimators neglect observational noise, resulting in biased estimates being obtained. Therefore, there is an essential need to determine more robust estimators.

By analyzing the resulting interaction surface topology together with the resulting interaction network structure, the following can be concluded: an early-warning sign of a potential crisis can be detected in the long run, relatively, as a consequence of capturing critical transitions. Such transitions are able to imply unwanted collapse. In addition, the reconstruction of the interaction networks is helpful in distinguishing between the markets which are most strongly affected by the crisis. This analysis, in turn, enables investors and market players to differentiate between those markets which will fall into a catastrophic loss and those which will not. As a consequence, they can make successful decisions towards selecting less risky portfolios, which guarantees lower losses.

In summary, various algorithms, as well as DPC techniques, can be applied to international stock-market time-series data for the purpose of reconstructing the time-dependent stock-market interaction parameter spaces. This indicates how the topology of the resultant interaction parameter spaces may change from a non-crisis state into a state in which a crisis occurs. Furthermore, DPC analysis provides a clear picture of how markets interact. This gives a clear warning sign which confirms the tendency for a potential crisis. This, in turn, allows investors to manage their losses, before their occurrence.

The remainder of this manuscript is structured as follows. The methods applied in this work are presented in Section 2. In Section 3, the applications of the methods to American stock markets are discussed.

## 2. Methods

This section presents the methods used in this manuscript, which are applied to American stock-market time-series data. The work is split into two parts, according to the aim of the manuscript. The first part introduces the methods used regarding the aim of constructing the interaction parameter spaces as density heat maps, in order to track the market motions through their topological changes during different periods of time. For this aim, the ARE algorithm is used. A diagram illustrating the underlying mechanism of the EM-KF algorithm is shown in Figure 1. The second part presents the method used for the aim of reconstructing the interaction network structure between the strongly interconnected market indices (i.e., DPC). For simplicity, an illustrative diagram of the methodology utilized in the manuscript is shown, step-wise, in Figure 2.

In the following, the SSM model is presented in the first sub-section. In the second sub-section, the model order-selection criterion $AIC_{i}$ is detailed. The EM-KF scheme is discussed and illustrated in the third sub-section. Finally, the DPC technique is presented in the fourth sub-section.

### 2.1. State Space Model (SSM)

The state space model (SSM) is a method for modeling both observed and hidden processes in a given system. The SSM model is used in the Kalman filter (KF) to model the data under analysis. This model contains two equations. The first equation models the dynamics or the state of the process and the Gaussian distributed driving noise, and it is called the state equation. The second equation models the observations with Gaussian distributed observational noise, and is called the output equation [26].

The dynamics of the underlying process are modeled by a linear stochastic equation, which is the vector autoregressive process of order *p* (VAR[*p*])
(1)$$x_{t}=\sum_{\tau =1}^{p}A_{t\tau }x_{t-\tau }+\epsilon_{t},\;\;\;\;\;\;\epsilon_{t}∼N\left(0,\;Q\right),$$
where $x_{t}$ is the current state vector based on the past *p* state vector and the Gaussian driving noise $\epsilon_{t}$, with zero mean and co-variance matrix *Q*. The transition matrix A(τ) varies over time, which, in turn, determines the dynamics of the process. The state and noise vectors $x_{t}$ and $\epsilon_{t}$ are of dimension *d*, while *A* and *Q* are d×d matrices.

On the other hand, the observation
(2)$$y_{t}=C\;x_{t}+\epsilon_{t},\;\;\;\;\;\;\eta_{t}∼N\left(0,\;R\right),$$
is modelled by the b×pd observation matrix *C* together with the observational noise $\eta_{t}$. The dimension of $y_{t}$ and $\eta_{t}$ is b×1. The observational noise is assumed to be Gaussian-distributed with zero mean and b×b co-variance matrix *R*. In general, b≠pd is where not all components of the underlying process are observed. This is a special case when reformulating a VAR[*p*] as a VAR[1], where the first *d* hidden states are only observed. Then, the state space model (SSM) can be written as [26,31]
(3)$$\begin{array}{ccc} x_{t} & = & A_{t}\;x_{t-1}+\epsilon \left(t\right)\\ y_{t} & = & C\;x_{t}+\eta \left(t\right) \end{array}$$

To sum up, the SSM consists of two equations: an equation that describes the dynamics of the process, in addition to an observation equation that models the observation function, and observational noise.

### 2.2. Model Order Selection Criterion AIC_i_

In time-series applications, before conducting the analysis, an appropriate model order must be chosen to characterize the collected data. The most standard criterion of scientific theory for this determination is the so-called Akaike information criterion (AIC), introduced by Akaike in 1974 [64]. This criterion is considered a data-driven selection method. The AIC can be obtained by evaluating:(4)$$AIC=-2logf(y|{\hat{\Theta }}_{k})+2k,$$
where $\hat{\Theta_{k}}$ denotes the parameter estimate that is obtained by maximizing the likelihood function for the model, $f(y|\Theta_{k})$ is the maximum, and *k* is the number of estimated parameters. Therefore, $f(y|\hat{\Theta_{k}})$ represents the resulting empirical likelihood.

In addition, the AIC provides insight into how a fitted model is close to the underlying generating (or true) model; this approach might suit some models, but not all. To this end, based on extending the original work of Akaike, Sugiura (1978) [65] proposed the AIC${}_{c}$, which is a corrected version of AIC developed in the context of regression models with normal errors. In such a setting, the AIC${}_{c}$ can be obtained by evaluating:(5)$$AIC_{c}=-2logf\left(y|{\hat{\Theta }}_{k}\right)+\frac{2Tk}{T-k-1},$$
where $\frac{2Tk}{T-k-1}$ is the bias correction term and *T* is the sample size. However, the effectiveness of the AIC${}_{c}$ motivates the need for an improved variant of AIC for state-space models, as has been demonstrated in [66]. This variant is based on an idea presented by Hurvich, Shumway, and Tasai (1990) [67], in the context of autoregressive models. This model is known as the “improved" Akaike information criterion (AIC${}_{i}$), which can be obtained by evaluating:(6)$$AIC_{i}=-2logf\left(y|{\hat{\Theta }}_{k}\right)+{\hat{B}}_{T}\left(k,\Theta_{s}\right),$$
where the penalty term ${\hat{B}}_{T}\left(k,\Theta_{s}\right)$ serves as a Monte-Carlo approximation [67,68].

The development of the AIC${}_{i}$ for state-space applications, as well as its performance, have been investigated in [68] through a simulation study. In addition, they compared the performance of the AIC, AIC${}_{c}$, AIC${}_{i}$, and other criteria, and found that the AIC${}_{i}$ outperformed the others in the context of SSM, where it provides the true model order.

As the AIC${}_{i}$ is utilized in this manuscript, then the log returns are used. The price of an asset at time 0 is denoted by $P_{0}$, and the price of an asset at time T is denoted by $P_{T}$. The log-return formula is given by:(7)$$\ln\frac{P_{T}}{P_{0}}.$$

To this end, in this manuscript, for state-space models, the AIC${}_{i}$ is utilized to match the requirements of such models and provide a more accurate model-order selection.

### 2.3. Expectation-Maximisation (EM) Algorithm and Kalman Filter (KL)

In this subsection, for the estimation of the state-space model parameters, expectation-maximisation (EM) algorithm and Kalman filter are utilized and presented [27,69,70]. The expectation-maximisation (EM) algorithm is based on an iterative scheme which consists of two steps: the expectation step and the maximization step, accordingly (see Figure 1). In the expectation step, conditional expected values of the hidden state x(t) and its covariance P(t) are obtained using the Kalman filter based on the equations explained above. In the maximization step, based on these values, the expected value of the likelihood is maximized with respect to the parameters, which results in a new set of parameters, which is used in the next iteration of the EM algorithm [26]. In the first iteration of the EM algorithm, the parameters P(1) need to be initialized. Therefore, for instance, the least-squares parameter estimates can then be used.

In other words, the expectation-maximization (EM) algorithm provides an iterative maximum likelihood estimator for the parameters in the state-space model (SSM) [28,30]. This EM algorithm for SSM is based on the so-called Kalman filter [29]. This filter is utilized to obtain estimates of the hidden states. The state estimates are then used to improve the estimates of the process parameters [26,27].

To introduce the Kalman filter, a measurement time series containing *n* observations is assumed with time t=1,⋯,n is used to reference these observations [26]. For conditional expectations [31]
(8)$$\begin{array}{ccc} x_{t}^{s} & = & E[x_{t}|y_{1},\cdots ,y_{s}], \end{array}$$
(9)$$\begin{array}{ccc} P_{t_{1},t_{2}}^{s} & = & E[\left(x_{t_{1}}-x_{t_{1}}^{s}\right){\left(x_{t_{2}}-x_{t_{2}}^{s}\right)}^{T}]\\ & = & E[\left(x_{t_{1}}-x_{t_{1}}^{s}\right){\left(x_{t_{2}}-x_{t_{2}}^{s}\right)}^{T}|y_{1},\cdots ,y_{s}]. \end{array}$$

The subscript denotes the estimation time point, while the superscript is up to which measurement it is conditioned on. The equality in Equation (Equation 8) holds if the underlying process is Gaussian, which is assumed here. The Kalman filter is described in terms of a set of equations which are based on an effective recursive computational way to estimate the state of the SSM process, which minimizes the mean of the squared error [26]. The Kalman-filter equations are [71]
(10)$$\begin{array}{ccc} x_{t}^{t-1} & = & A\;x_{t-1}^{t-1} \end{array}$$
(11)$$\begin{array}{ccc} x_{t}^{t} & = & x_{t}^{t-1}+K_{t}\;\left(y_{t}-C\;x_{t}^{t-1}\right) \end{array}$$
(12)$$\begin{array}{ccc} P_{t}^{t-1} & = & A\;P_{t-1}^{t-1}\;A^{T}+Q \end{array}$$
(13)$$\begin{array}{ccc} P_{t}^{t} & = & P_{t}^{t-1}-K_{t}\;C\;P_{t}^{t-1} \end{array}$$
(14)$$\begin{array}{ccc} K_{t} & = & P_{t}^{t-1}\;C^{T}\;{\left(C\;P_{t}^{t-1}\;C^{T}+R\right)}^{-1}, \end{array}$$
with initial values $x_{0}^{0}=E\left[x_{t}\right]=\mu$ and $P_{0}^{0}=E\left[x_{t}\cdot x_{t}^{T}\right]=\Sigma$. The idea of how the Kalman filter works is based on a recursive cycle of a time-update and a measurement-update step [72]. The time update, in Equations (Equation 10) and (Equation 12), predicts the state from time t−1 to *t*, which results in the prior estimate $x_{t}^{t-1}$ and its co-variance $P_{t}^{t-1}$[26]. The measurement-update step consists of Equations (Equation 11) and (Equation 13), and it corrects the prior estimates by taking into account the current prediction $x_{t}^{t-1}$, the measurement $y_{t}$ and the Kalman-filter gain, in Equation (Equation 14); this leads to the posterior estimates [26]. The Kalman-filter equations apply a recursive scheme, as only the observations and estimates from the past and the present are used [73]. Applying the steps of the EM algorithm together with the Kalman filter iteratively ensures convergence to the best estimator of the underlying dynamics and the parameters of the process [27].

Maximum-likelihood estimation (MLE) is one of the most effective approaches to fit model parameters to data. The likelihood is a function which describes the probability of the recorded data given the model parameters. The maximization of the likelihood function results in obtaining the parameters of the model where the observed time series is most likely. An iterative maximum-likelihood estimator of the SSM parameters is derived [26]. For the complete data log-likelihood:(15)$$\begin{array}{ccc} log\;L= & - & \frac{1}{2}log\left|\sum \right|-\frac{1}{2}{\left(x_{0}-\mu \right)}^{T}{\left(\sum \right)}^{-1}\left(x_{0}-\mu \right)\\ & - & \frac{n}{2}log\left|Q\right|-\frac{1}{2}\sum_{t=1}^{n}{\left(x_{t}-Ax_{t-1}\right)}^{T}Q^{-1}\left(x_{t}-Ax_{t-1}\right)\\ & - & \frac{n}{2}log\left|R\right|-\frac{1}{2}\sum_{t=1}^{n}{\left(y_{t}-Cx_{t-1}\right)}^{T}R^{-1}\left(y_{t}-Cx_{t}\right). \end{array}$$

Since the hidden states $x_{t}$ are unknown, only the expected value
(16)$$\begin{array}{ccc} G(\Theta )= & - & E(logL|y_{1},...,y_{n})\\ & = & -\frac{1}{2}log\left|\sum \right|-\frac{1}{2}tr\left\{\sum^{-1}\left(P_{0}^{n}+\left(x_{0}^{n}-\mu \right){\left(x_{0}^{n}-\mu \right)}^{T}\right)\right\}\\ & - & \frac{n}{2}log\left|Q\right|-\frac{1}{2}tr\left\{Q^{-1}\left(F-EA^{T}-AE^{T}+ADA^{T}\right)\right\}\\ & - & \frac{n}{2}log\left|R\right|\\ & - & -\frac{1}{2}tr\left\{R^{-1}\sum_{t=1}^{n}\left[\left(y_{t}-Cx_{t}^{n}\right){\left(y_{t}-Cx_{t}^{n}\right)}^{T}+CP_{t}^{n}C^{T}\right]\right\} \end{array}$$
of the log-likelihood conditioned on $y_{1},...,y_{n}$ is accessible. The illustration of the abbreviations used in the previous equation is as follows:(17)$$\begin{array}{ccc} D & = & \sum_{t=1}^{n}\left(P_{t-1}^{n}+x_{t-1}^{n}{x_{t-1}^{n}}^{T}\right),\\ E & = & \sum_{t=1}^{n}\left(P_{t,t-1}^{n}+x_{t}^{n}{x_{t-1}^{n}}^{T}\right),\\ F & = & \sum_{t=1}^{n}\left(P_{t}^{n}+x_{t}^{n}{x_{t}^{n}}^{T}\right). \end{array}$$

The quantities required in Equation (Equation 17) are the results of the Kalman filter of the r-th EM iteration. To maximize G(Θ), its derivative is set to zero, leading to the update rules:(18)$$\begin{array}{ccc} A^{r+1} & = & ED^{-1} \end{array}$$(19)$$\begin{array}{ccc} Q^{r+1} & = & \frac{1}{n}\left(F-ED^{-1}E^{T}\right), \end{array}$$(20)$$\begin{array}{ccc} R^{r+1} & = & \frac{1}{n}\sum_{t=1}^{n}\left[\left(y_{t}-Cx_{t}^{n}\right){\left(y_{t}-Cx_{t}^{n}\right)}^{T}+CP_{t}^{n}C^{T}\right]\}. \end{array}$$

The update of μ is $x_{0}^{n}$ of the last iteration of EM. If the measurement is corrected for the mean, then the initial value for the first EM iteration of μ is set to zero. In addition, the initial value of the co-variance of the process ∑ can be estimated or set to a reasonable baseline value [71]. In addition, the likelihood never decreases; therefore, there is no adjustment of step size is needed [28].

In summary, applying the EM algorithm together with the Kalman filter is a robust iterative procedure to estimate model parameters in the SSM, in addition to de-noising the time-series data. The main drawback of this approach is that it has high computational burden.

### 2.4. Granger Causality in the Time Domain: Directed Partial Correlation (DPC) [74]

In order to provide a time-domain measure based on the concept of Granger causality, the directed partial correlation (DPC) was introduced by Eichler (2005) [56]. One of the most effective features of DPC is that it can be used as a measure of causal-effect strength [56]. When inferring causal relationships from time-series data, VAR[*p*] models can be fitted using least-squares estimation [75], which is utilized in this manuscript. For observations $x_{V}\left(1\right),\cdots ,x_{V}\left(T\right)$ from a *d*-dimensional multiple time series $x_{V}$, we obtain a vector autoregressive model (VAR) with the following representation:(21)$$x_{V}\left(t\right)=\sum_{r-1}^{P}\;A\left(r\right)\;x_{V}\left(t-r\right)\;+\epsilon_{V}\left(t\right),$$
where $x_{V}\left(t\right)$ is the vector that represents the entire set of observed processes. Now, let ${\hat{R}}_{p}={\left({\hat{R}}_{p}\left(h,\nu \right)\right)}_{h,\nu =1,\cdots ,p}$ be the pd×pd matrix composed by sub-matrices [56]
(22)$${\hat{R}}_{p}\left(h,\nu \right)=\frac{1}{T-p}\;\sum_{t=p+1}^{T}\;x\left(t-h\right)\;x{\left(t-\nu \right)}^{T}.$$

Similarly, ${\hat{r}}_{p}$ is set to be such that ${\hat{r}}_{p}=\left({\hat{R}}_{p}\left(0,1\right),\cdots ,{\hat{R}}_{p}\left(0,p\right)\right)$. Then, the least-squares estimates of the autoregressive coefficients are given by
(23)$$\hat{A}\left(h\right)=\sum_{\nu =1}^{p}\;{\left({\hat{R}}_{p}\right)}^{-1}\left(h,\nu \right)\;{\hat{r}}_{p}\left(\nu \right),$$
where h=1,⋯,p and the covariance matrix Σ is estimated by
(24)$$\hat{\Sigma }=\frac{1}{T}\;\sum_{t=p+1}^{T}\;\hat{\varepsilon }\left(t\right)\;\hat{\varepsilon }{\left(t\right)}^{T},$$
where
(25)$$\hat{\varepsilon }\left(t\right)=x\left(t\right)-\sum_{h=1}^{p}\;A\left(h\right)\;x\left(t-h\right)$$
are the least-squares residuals. However, the coefficients $A_{ij}\left(h\right)$ depend on the unit of measurement of $x_{i}$ and $x_{j}$; thus, they are not suitable for comparisons of the strength of causal relationships between variables [56]. Therefore, Eichler (2005) [56] proposed DPC as a measure of the strength of causal relationships. For h>0, the DPC $\pi_{ij}\left(h\right)$ is defined as the correlation between $x_{i}\left(t\right)$ and $x_{j}\left(t-h\right)$ after removing the linear effects of the other variables included in the vector $x_{V}$. For h<0, $\pi_{ij}\left(h\right)=\pi_{ij}\left(-h\right)$. In addition, it has been shown in [56] that for h>0, estimates for the DPC $\pi_{ij}\left(h\right)$ can be obtained from the parameter estimates of a VAR[*p*] model by re-scaling the coefficients $A_{ij}\left(h\right)$
(26)$${\hat{\pi }}_{ij}\left(h\right)=\frac{{\hat{A}}_{ij}\left(h\right)}{\sqrt{{\hat{\Sigma }}_{ii}\;{\hat{\rho }}_{jj}\left(h\right)}}\;\;\mathrm{for}\;\;j\to i\;,$$
where
(27)$${\hat{\rho }}_{jj}\left(h\right)={\hat{K}}_{jj}+\sum_{\nu =1}^{h-1}\;\sum_{k,l\in V}\;{\hat{A}}_{kj}\left(\nu \right)\;{\hat{K}}_{kl}\;{\hat{A}}_{lj}\left(\nu \right)+\frac{{\hat{A}}_{ij}{\left(h\right)}^{2}}{{\hat{\Sigma }}_{ii}}.$$

The matrix $\hat{K}={\hat{\Sigma }}^{-1}$ is the inverse of the estimated covariance matrix $\hat{\Sigma }$ of the residual noise processes.

To decide the significance of an estimated causal influence, we use a statistical evaluation procedure based on bootstrapping to construct the confidence interval as follows [74]:Generate *B* bootstrap surrogates (resamples) with the same length as the original data. A rough minimum of 1000 bootstrap surrogates is often sufficient to compute accurate confidence intervals, as has been suggested by Efron and Tibshirani [76]. Here, *B* is set to 10,000. The surrogates are generated using a non-parametric method—the amplitude-adjusted Fourier transform (AAFT) which was originally proposed by Theiler et al. (1992) [77,78]. This method works under the null hypothesis that the original data are generated from a stationary, Gaussian and linear stochastic process [79]. The algorithm for generating the surrogates is described as follows [79,80]:(a)The original data are re-scaled to a normal distribution. This is based on a simple rank ordering, which is performed by generating a time series with Gaussian distribution which is then sorted according to the original data.(b)A Fourier-transformed surrogate of the re-scaled data is constructed.(c)The final surrogate is scaled to the distribution of the original data by sorting the original data to the ranking of the Fourier-transformed surrogate.The use of this algorithm is advantageous as it preserves the distribution, as well as approximately preserving the power spectrum (i.e., the autocorrelation structure), of the original data [79,80]. For the implementation of the AAFT method, we used the Tisean package (for details about the Tisean package, we refer to http://www.mpipks-dresden.mpg.de/tisean/) [78]. Note that the Tisean program performs the algorithm described above, iteratively, until no further improvement can be made [78].Estimate the DPC for each *B* bootstrap surrogates to yield a bootstrap sampling distribution, i.e. ${\left\{{{\hat{\tau }}_{r}}^{★}\right\}}_{r=1,\cdots ,B}$. To obtain the (1−α)100 percentile bootstrap confidence interval for $\hat{\tau }$, the sampling distribution values of ${{\hat{\tau }}_{r}}^{★}$ are sorted in ascending order. Then, the α percent and (1−α) percent points are chosen as the end points of the confidence interval, giving [${{\hat{\tau }}_{r}}^{★}\left(\alpha B\right),{{\hat{\tau }}_{r}}^{★}\left(\left(1-\alpha \right)B\right)$] [81]. For a 95% confidence interval with B=10,000, this would be approximately [${\hat{\tau }}^{★}\left(500\right),{\hat{\tau }}^{★}\left(9500\right)$].If the DPC value estimated from the original time series lies outside the confidence interval, then the value is considered to be significantly different from zero.

### 2.5. Degree-Centrality Measures

In this subsection, degree-centrality measures are described. Degree centrality corresponds to the total number of connections linked to a node of a network [82]. Degree centrality has two measures: mainly in-degree and out-degree. In-degree refers to the number of connections that point inward at a node, while out-degree refers to the number of connections that originate at a node and point outward to other nodes [83]. In this manuscript, the use of these measures is considered advantageous. The in-degree measure identifies the most affected market indices, while the out-degree measure identifies the most influential market indices. This differentiation is crucial for investors and market players in the decision-making process related to investment portfolios.

## 3. Application to American Stock Markets—Subprime Mortgage Crisis (2007–2008)

This section presents the results of applying the ARE algorithm to American stock markets. American stock-market time-series data are introduced in the first part of this section. Before estimating the parameters, the model order must be obtained. To this end, a model order selection criterion is utilized in the second part of this section. The final part of this section presents the results and our conclusions.

### 3.1. Data

The data sets included of 41 American stock-market indices for 14 countries. Therefore, the sample size was 41 data sets, each of which had 1417 observations. The indices for the markets of respective countries are displayed in Table A1. The data were collected from the Yahoo Finance database, on the basis of daily closing prices [84]. The analysis covered the whole period of years 2006–2010 which were divided into 5 periods, namely,

(a) 1/1/2006 to 30/6/2006 (first half of 2006)(b) 1/7/2006 to 30/6/2007 (second half of 2006 to first half of 2007)(c) 1/7/2007 to 31/12/2007 (second half of 2007)(d) 1/1/2008 to 31/12/2008 (2008)(e) 1/1/2009 to 31/12/2010 (2009–2010)

### 3.2. Model-Order Selection Criterion AIC${}_{i}$

In this subsection, the results of employing the AIC${}_{i}$ criterion to calculate the SSM model order are presented in Table 1. It can be seen that the true model order corresponded to the largest AIC${}_{i}$ value, which means that the optimal chosen order was three for the estimation process of the autoregressive coefficients for the SSM Model. Knowing the true model order enables accurate estimation of the autoregressive coefficients, (i.e., $\alpha_{1},\alpha_{2}$, and $\alpha_{3}$) of the SSM by EM-KF scheme. These three autoregressive coefficients were estimated for each time period studied.

### 3.3. Results

The numerical algorithm ARE was applied to the 41 stock-market time-series data sets. The main objective of utilizing the ARE algorithm was to observe the pattern and the tendency of the market’s movements, in order to distinguish between different crisis states. In other words, the focus of the ARE algorithm was to observe the general pattern of the market flow and how the markets move from one state to another over time (2006–2010). This allows for tracking market motions, for the purpose of early-warning detection of any unusual specific pattern. This tool is ideal for knowledge discovery in data sets, as it determines the grouping structure in time-series data [26,27,35,69]. The ARE analysis was conducted for each of the above-mentioned periods (detailed in Section 3.1 separately, with no overlap between them, in order to demonstrate how the topology of the constructed interaction surfaces of the stock markets under study changed from one state to another. Furthermore, the DPC technique was further employed to identify the most affected markets, as well as to determine the entire causal interaction structure.

In the following, the discussion of each reconstructed interaction space, as well as the corresponding causal interaction structure, is presented. Note that the main interest of conducting DPC analysis was to draw conclusions regarding the interaction structures among the most affected markets, which are strongly interconnected. As only strong interactions were of interest here, only the interconnectivity links which were larger than or equal to 0.65 are shown. For the ARE constructed surface, each interaction surface was constructed based on the three leading estimated autoregressive parameters (i.e., $\alpha_{1},\alpha_{2}$, and $\alpha_{3}$). These estimated parameters are the coordinates of each point, which corresponds to each stock market; that is, for every point in time. The three-dimensional parameter spaces are shown as snapshots (i.e., time frames) representing the motion and the behavior of the markets over each period separately. More precisely, the estimation process resulted in a sequence of different sets of parameter values describing the state of each point, which represents each market in the parameter space. For a smoother view of the constructed surfaces, they are presented as heat maps, according to the density reflecting the interaction levels among markets.

The results of conducting the ARE algorithm, when there was no crisis, are presented in Figure 3, which shows the inferred interaction parameter space during the time period 1/1/2006 to 30/6/2006. This space was reconstructed based on the three auto-correlation coefficients estimated using the EM-KF scheme. This estimate determines the coordinates of the position of each market. This, in turn, provides the pattern of the market’s movements. The color bar shows the heat map, representing the density where markets are positioned in the same place. In other words, the color becoming more red, reflects higher density. As such, the strong interactions among markets are found only in the yellow-red regions presented in Figure 3, while the low and the medium interactions are found in the blue regions. Finally, the white regions represent no interaction.

On the other hand, in order to identify which markets were strongly interacting, DPC analysis was conducted. Figure 4 and Figure 5 show the inferred interaction network structures corresponding to the two yellow-red regions in Figure 3. Figure 4 reflects the underlying constructed interaction network structure corresponding to the first yellow-red region, which is located on the lower left side of the surface in Figure 3, while Figure 5 reflects the underlying constructed interaction network structure corresponding to the second yellow-red region, which is located on the top-right side of the surface in Figure 3. In Figure 4 and Figure 5, the color of the nodes corresponds to the country indices (see Table A1 in the Appendix A). In addition, the thickness of the arrows refers to the strength of interactions among markets. Note that, in this study, we focus only on interaction parameters equal to 0.65 and above, which reflect the strongest interactions. Each node represents the name of the market index. Here, there were four U.S. markets (nodes 3, 4, 5 and 6), and one Panama market (node 2) and one Canada market (node 1). According to Figure 4, which represents Figure 3, region 2, on the one hand, there was a strong interaction between U.S. market indices. This formed a community of strongly interacting U.S. market indices with an influence on one of Brazil’s market indices. It can be observed that the link (8 (Brazil) →5 (U.S.)) was present only due to the strong interaction among U.S. markets which, in turn, affects Brazil. This led to Brazil influencing one of the U.S. markets in return. On the other hand, Figure 5, which represents Figure 3, region 2, demonstrates that the U.S. market indices strongly influenced both Panama and Canada markets. The link (2 (Panama) →1 (Canada)) is present as a result of the strong influence of U.S. market indices on node 2 (Panama). The same situation occurs for the link (2 (Panama) →6 (U.S.)), as this link is present due to the strong influence of U.S. markets on Panama markets. This also occurred for the link (1 (Canada) →6 (U.S.)), appearing as a consequence of the strong influence of the U.S. market indices on node 1 (Canada).

In general, for the first time period, it can be observed that a small number of the markets were strongly interconnected, where the rest moved in a distributed manner over the constructed surface. In particular, the interaction surface formed two small communities of markets which were very close to each other.

For the second time period (1 July 2006 to 30 June 2007), ARE and DPC analyses were also conducted. The ARE results are presented in Figure 6, and it can be seen that almost all markets were settled in one particular region with high density. In addition, there were a small number of markets that did not belong to the high-density cluster. Furthermore, it can be seen that the density of the collective motion becomes lower in the middle of the surface and almost zero at the end of it. This indicates a special pattern that occurs, which can be considered a warning sign regarding a crisis that will happen at some point in the future. Furthermore, to identify which markets are the most strongly interconnected, a DPC analysis was conducted. The inferred interaction network structure that corresponds to the yellow-red region is presented in Figure 7. The strong interconnectivity structure among U.S. markets was clearly detected. The reason behind the appearance of links going out to nodes 13 (Brazil), 14 (Brazil), 15 (Canada), 16 (Canada), and 17 (Colombia) was the strong influence of all U.S. market indices on Brazilian, Canadian, and Colombian markets. This indicates that the Brazilian, Canadian, and Colombian markets will potentially be the most affected markets due to the U.S. home mortgage crisis. This conclusion is evidenced in Figure 10.

For the third time period (1/7/2007 to 31/12/2007), Figure 8 presents the reconstructed surface, where the high density of the markets is moving collectively from one state to another. The behavior direction is indicated by an arrow. Interestingly, this collective motion is known as “herding behavior” in the literature [85,86]. In order to identify the markets which are collectively moving, a DPC analysis was conducted, and the corresponding interaction network structure is reconstructed (see Figure 9). To distinguish between the most and least affected markets, the degree centrality measure was utilized. Table 2 provides the results for the calculation of the out-degree and the in-degree of each node presented in Figure 9. According to Table 2, the most influencing nodes were 1, 5, 10, and 16, while nodes 3 and 8 were the most affected markets.

The analysis for the fourth time period (1 January 2008 to 31 December 2008), the state where the crisis broke out and reached its peak (in 2008), is shown in Figure 10. It can be observed that most companies moved to a different state, except a few of them. The figure also shows that a high density of markets settled at another position on the right side of the surface. This illustrates that the high-density cluster contained the vast majority of the markets which were strongly interconnected with each other. In order to distinguish the most and the least affected markets, a DPC analysis was conducted; the result is shown in Figure 11, which shows that the markets which strongly interacted before the crisis happened (see Figure 6) are those that are affected here, forming a cluster (see Figure 10). It can be observed that nodes 3 and 6 were both influenced by nodes 5 and 11, while nodes 7 and 15 influenced nodes 3 and 6. In addition, node 16 (Canada) was seen to influence the other markets; namely, nodes 9 (U.S.), 15 (Canada), and 17 (Colombia). To identify the most central influencing market indices, the degree-centrality measure was applied. The out-degree and in-degree for each node was calculated, and the results are given in Table 3. According to Table 3 corresponding to Figure 11, it can be noted that nodes 1, 5, and 10 were the most influencing U.S. markets on all the other markets. Furthermore, node 8 (U.S.) was the most affected U.S. market during the crisis, which influenced node 15 (Canada) through nodes 2 and 4. In addition, node 3 (U.S.) was the second-most affected U.S. market, which transmitted the crisis into Brazil (node 14) via node 10 (U.S.); which, in turn, affected node 13 (Brazil).

In summary, the crisis was transmitted from U.S. markets to Brazil markets, which were the most affected markets during the crisis, followed by Canada and (the least affected) Colombia.

For the final time period (1 January 2009 to 31 December 2010), after the crisis had finished, the surface returned to a state where no obvious pattern could be captured, and the markets were distributed all over the surface again (see Figure 12).

Figure 12 shows that the topological structure of the surface is changed and formed into different clusters. To identify these clusters, the reconstructed interaction network structure based on DPC is presented in Figure 13. The first cluster contains nodes 2, 3, 7, and 9 (U.S. market indices), the second cluster contains nodes 5 and 6 (Canada market indices), the third cluster contains nodes 12 and 13 (Brazil market indices), and the final cluster contains nodes 1, 4, 8, 10, and 11 (U.S. market indices).

In comparison, Figure 6 and Figure 10 show that all the markets were entirely connected to each other, forming obvious clusters, in contrast to the connectivity structure presented in Figure 12. Based on this connectivity structure, no specific pattern could be captured. However, U.S. markets continued their influence on Brazilian and Canadian markets after the crisis.

Based on the observed interaction pattern, as well as its corresponding structure in Figure 7 and Figure 11, the following can be concluded. The market indices observed before the crisis, toward the crisis, and during the crisis were the same for the three phases, as confirmed by the same clustering pattern being observed in these Figures.

In summary, the states in which the markets were not falling into a crisis or where no potential crisis existed are shown in Figure 3 and Figure 12.

Interestingly, in the comparison between Table 2 referring to Figure 9 with Table 3 referring to Figure 11, it can be determined that the most affected markets corresponded to nodes 3 and 8. This conclusion indicates the importance of conducting DPC analysis together with calculating out-degree and in-degree measures, in order to provide a warning sign and identify which markets may be the most affected.

To sum up, an illustrative graph for showing the transition between the time before the crisis-period (b)-(second half of 2006 to first half of 2007) and the time during the crisis-period (d)-(2008), is presented in Figure 14.

## 4. Application to American Stock Markets: COVID-19 (2020)

To show the robustness of the methodology presented in this manuscript, in this section further analysis is performed to cover one more crisis. We take the COVID-19 outbreak as another example to conduct the same analysis.

### 4.1. Data

The data sets utilized in this section are the same 41 American stock-market indices for 14 countries. The data were collected from the Yahoo Finance database, on the basis of daily closing prices [84]. It is known in literature that the COVID-19 outbreak started on 20/2/2020 and reached the peak on 7/4/2020 [87,88,89,90,91,92,93]. For this reason, the analysis covered the period of years 2018–2021, which were divided into five periods, namely,

(a) No crisis: 1/10/2018 to 31/3/2019(b) Before crisis: 1/4/2019 to 31/10/2019(c) Towards crisis: 1/11/2019 to 28/2/2020(d) Crisis time: 1/3/2020 to 31/12/2020(e) After crisis: 2021

### 4.2. Model-Order Selection Criterion AIC${}_{i}$

In this subsection, the results of employing the AIC${}_{i}$ criterion to calculate the SSM model order are presented in Table 4. It can be seen that the true model order corresponded to the largest AIC${}_{i}$ value, which means that the optimal chosen order was three for the estimation process of the autoregressive coefficients for the SSM model. Knowing the true model order enables accurate estimation of the autoregressive coefficients, (i.e., $\alpha_{1},\alpha_{2}$, and $\alpha_{3}$) of the SSM by EM-KF scheme. These three autoregressive coefficients were estimated for each time period studied.

### 4.3. Results

The same methodology is employed for each period separately, to show the possibility of detecting a warning sign of a potential crisis, that is, in the framework of COVID-19. The results are presented in Figure 15. The figure shows that the surface topology for state (a) no crisis and state (e) after crisis have completely different topologies from the rest of surfaces. The surface presented in state (b) before crisis can be considered as a warning sign that there is a potential for crisis. The evidence has been shown in state (d) crisis time, when the cluster has moved from one state into another. The conclusion of the financial crisis resulting from the subprime mortgage crisis, presented in Section 3, can also be drawn for an early-stage detection of the financial crisis resulting from COVID-19 outbreak.

These results provide evidence that there is the possibility of detecting a warning sign some time before the actual crisis happens. Further analysis can be carried out by conducting DPC analysis similar to the one conducted in Section 3.

## 5. Discussion and Conclusions

The prediction of stock-market crashes has attracted interest over the years. Several researchers have studied this phenomenon using different approaches; however, the identification of which markets will be affected during the crisis has not been studied properly in the literature.

In this manuscript, the behavior of stock markets was demonstrated using the ARE algorithm. Based on the estimated SSM model order, the three-dimensional interaction parameter spaces were reconstructed, and a change in the topology of these spaces served to identify state transitions. Specifically, the EM-KF algorithm provides a means of constructing a space which shows how close each market is to others. When observing a cluster of markets smoothly presented as a heat map, a high density refers to strong interactions among markets. This, in turn, means that these markets may be the most strongly affected during a crisis. This approach provides an insight into the idea of collective motions of large numbers of entities. In other words, the use of this algorithm is beneficial and advantageous in the case of having big data, as it presents the results in the sense of pattern recognition.

For practical examples to validate the methodology introduced, two crises examples were studied. According to the analysis and results for both crises, there were two obvious state transitions; the first state refers to the time period before the crisis and the second state refers to the time period during the crisis. More precisely, the first state (before crisis) can be considered as a warning sign of a potential crisis. In addition, in the corresponding interaction parameter spaces, only the high-density regions were analyzed, in order to identify the most interacting markets specifically for the first state (before crisis). This means that both the most affected and most influential markets could be distinguished. To this end, DPC was utilized, such that interaction networks corresponding to these high-density regions could be reconstructed. As a first step, identifying the markets which could potentially succumb to a crisis is crucial. Furthermore, to distinguish between markets, the most affected and most influential markets, degree-centrality measures were used to calculate the in-degree and out-degree for each reconstructed network node.

These analyses results allow investors and market players to track those markets that are going through a potential crisis. In addition, it provides them a warning sign of the potential time that a crisis might occur. These results are expected to be of aid for investors, in terms of improving the decision-making process in portfolio selection. This allows them to reduce the risk exposure associated with their portfolios. Furthermore, investors can also exclude or withdraw their investments from companies which are expected to go through a potential crisis, in order to protect their investments against certain loss. To sum up, this methodology allows for early-stage detection of a financial crisis.

Such analysis can not only be carried out for financial markets, but also for other systems. For example, in neuroscience, recognizing certain patterns can provide early warnings for brain diseases, one of the main objectives in this field. Another example is the study of climatic changes to observe and detect certain patterns, which can be useful in predicting a potential catastrophe.

## Figures and Tables

**Figure 1 entropy-25-00070-f001:**
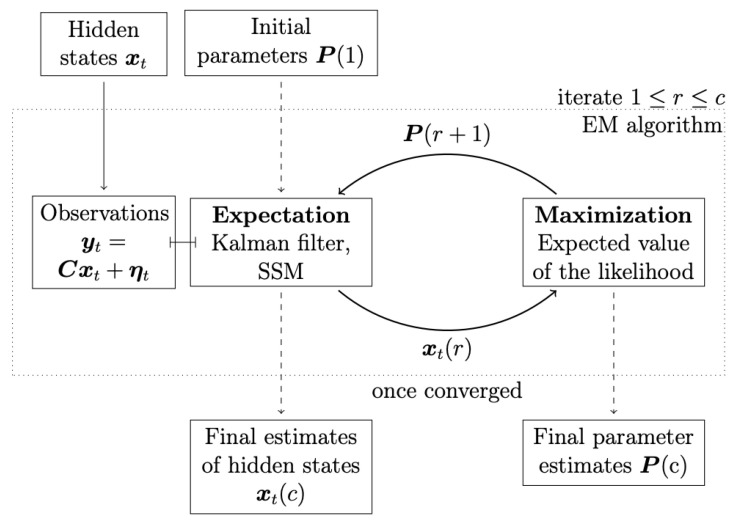
Kalman filter in the expectation-maximization algorithm. The Kalman filter is utilized to obtain conditional means using as parameters the P(r) in every iteration *r*. Maximization of the expected value of the likelihood function leads to a new set of parameters P(r+1) [26].

**Figure 2 entropy-25-00070-f002:**
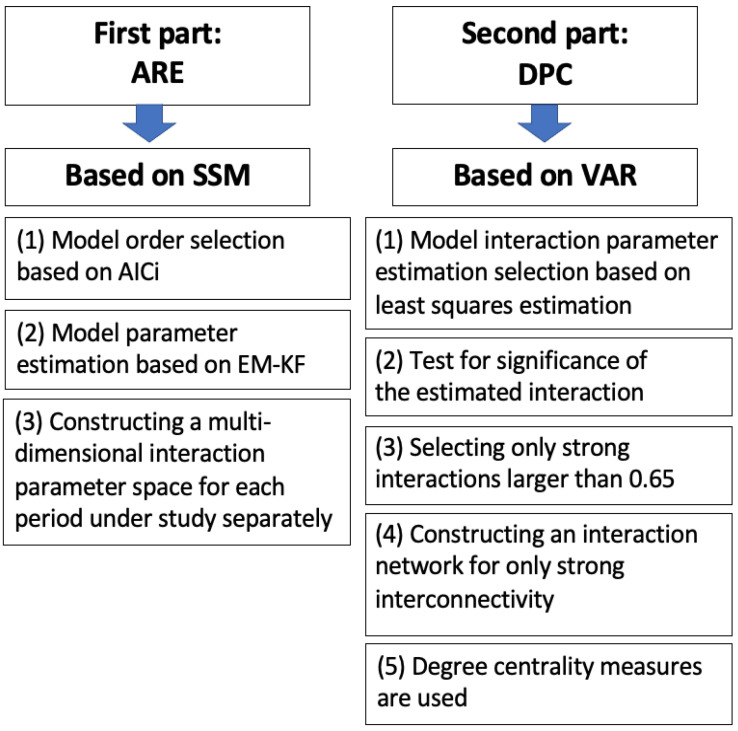
An illustrative diagram for the research methodology design utilized through the manuscript.

**Figure 3 entropy-25-00070-f003:**
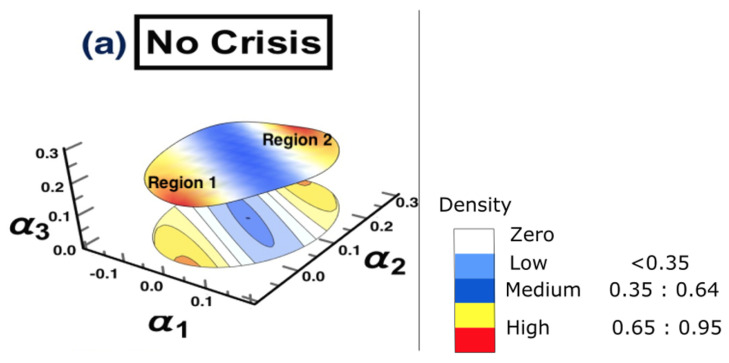
The constructed stock-market three-dimensional interaction parameter space, which corresponds to period (a) that represents the first half of 2006. This space was reconstructed based on the three estimated auto-correlation coefficients of the SSM model, where the estimate determines the coordinates of the position of each market. The figure demonstrates the level of interaction, which differs from one region to another. Note that the density bar is divided into three parts (low, medium, and high), with the corresponding interaction coefficients for each part. Therefore, the high-density spots correspond to high interaction among markets.

**Figure 4 entropy-25-00070-f004:**
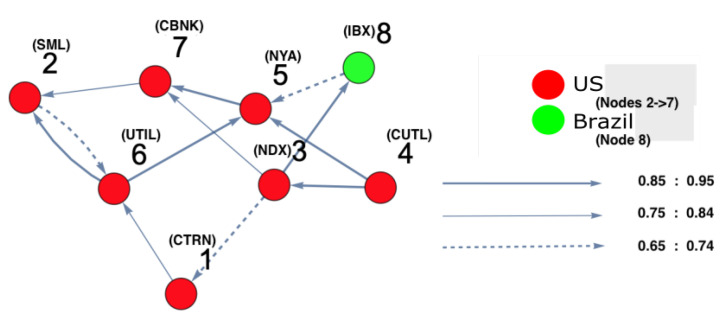
The constructed stock-market interaction network structure. The constructed network reflects the interaction structure among markets corresponding to *region 1* in the constructed space presented in Figure 3. The red-colored nodes correspond to U.S. stock markets, while the green-colored node corresponds to a stock market belonging to Brazil. The causal strength of interest to be represented in this manuscript is above 0.65, reflecting strong interactions. More precisely, three kinds of strongly connected causal links are presented here. The first are the dashed links, which correspond to a causal strength between 0.65 and 0.74; the second are the light-colored links, which corresponds to a causal strength between 0.75 and 0.84, the third are the dark-colored links, which correspond to a causal strength between 0.85 and 0.95. The network shows that the majority of connected indices mostly belong to U.S. markets.

**Figure 5 entropy-25-00070-f005:**
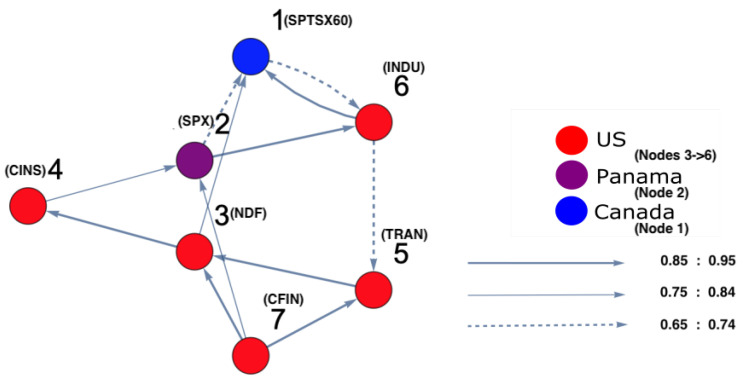
The constructed stock-market interaction network structure. The constructed network reflects the interaction structure among markets which corresponds to *region 2* in the constructed space presented in Figure 3. The red-coloured nodes correspond to U.S. stock markets, the blue-colored node corresponds to a stock market belonging to Canada and the purple-colored node corresponds to a stock market belonging to Panama. Recall that the causal strength of interest to be represented in this manuscript is above 0.65. The network shows that the strong connectivity structure is captured between Panamanian and Canadian markets with U.S. markets.

**Figure 6 entropy-25-00070-f006:**
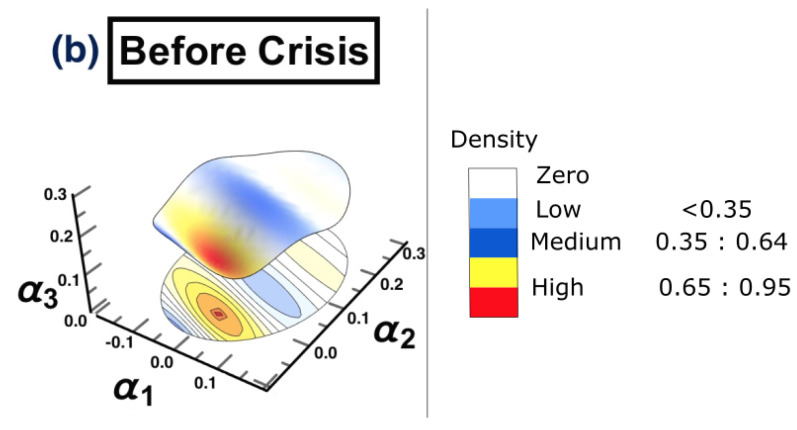
The constructed stock-market three-dimensional interaction parameter space, which corresponds to period (b) that represents the second half of 2006 to the first half of 2007. This space is reconstructed based on the three estimated auto-correlation coefficients of the SSM model, where the estimate determines the coordinates of the position of each market. The figure shows that there is a region where the density is very high, which refers to strong interactions among stock markets.

**Figure 7 entropy-25-00070-f007:**
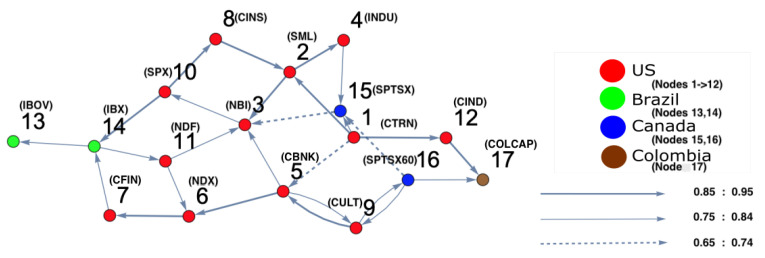
The constructed stock-market interaction network structure. The constructed network reflects the interaction structure among markets corresponding to the high-density region in the constructed space presented in Figure 6. The red-colored nodes correspond to U.S. stock markets, the blue-colored nodes correspond to stock markets belonging to Canada, the green-colored nodes correspond to stock markets belonging to Brazil and the brown-colored nodes correspond to stock markets belonging to Colombia. The network shows that nodes 14, 15, 16 and 17 are the most interacting market indices with U.S. markets.

**Figure 8 entropy-25-00070-f008:**
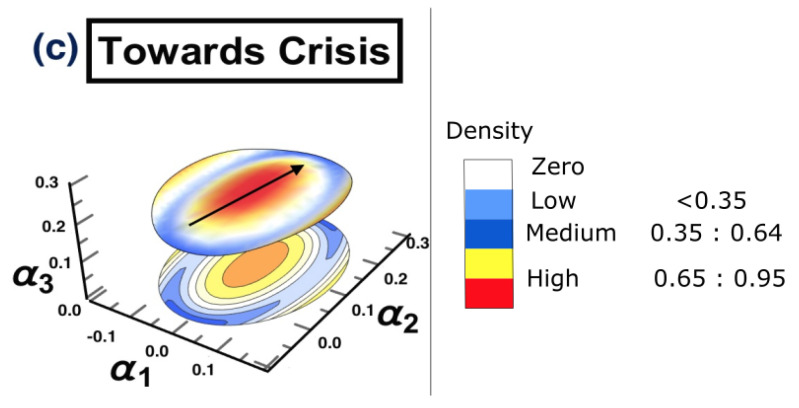
The topology of the constructed stock-market three-dimensional interaction parameter space, which corresponds to period (c) that represents the second half of 2007. Here, the whole market is in a state towards a crisis. The figure shows that there is a region in the middle of the space where the density is very high and wide, which is going in a specific direction, as indicated with an arrow.

**Figure 9 entropy-25-00070-f009:**
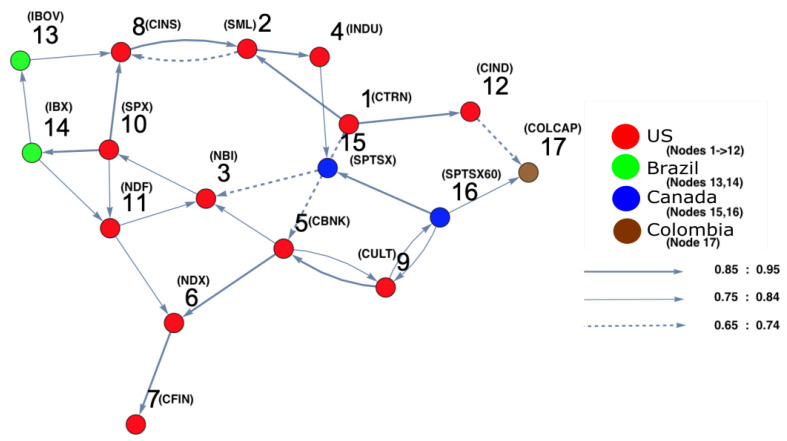
The constructed stock-market interaction network structure. The constructed network reflects the interaction structure among markets corresponding to the high-density region, which spans the red region along the directed arrow in the constructed space presented in Figure 8. The red-colored nodes correspond to U.S. stock markets, the blue-colored nodes correspond to stock markets belonging to Canada, the green-colored nodes correspond to stock markets belonging to Brazil and the brown-colored nodes correspond to stock markets belonging to Colombia. The network shows that all other markets are strongly interconnected with U.S. market indicies.

**Figure 10 entropy-25-00070-f010:**
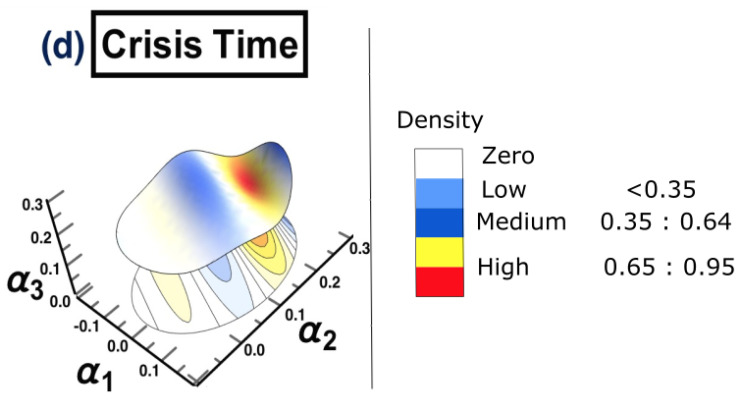
The constructed stock-market three-dimensional interaction parameter space, which corresponds to period (d) that represents 2008. The figure shows that there is a region where the density is very high, as it refers to strong interactions among stock markets which are falling into a crisis forming a cluster.

**Figure 11 entropy-25-00070-f011:**
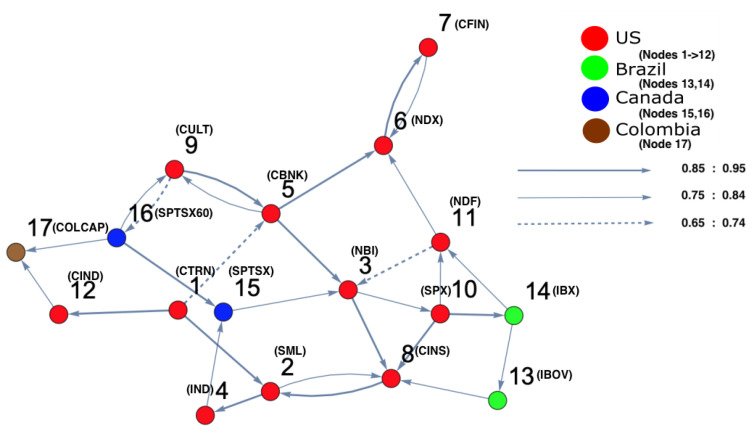
The constructed stock-market interaction network structure. The constructed network reflects the interaction structure among markets corresponding to the high-density region, which spans the red region along the directed arrow in the constructed space presented in Figure 10. The red-colored nodes correspond to U.S. stock markets, the blue-colored nodes correspond to stock markets belonging to Canada, the green-colored nodes corresponding to stock markets belonging to Brazil and the brown-colored nodes correspond to a stock market belonging to Colombia. The network shows that nodes 13, 14, 15 and 16 are the most interacting market indices with U.S. markets.

**Figure 12 entropy-25-00070-f012:**
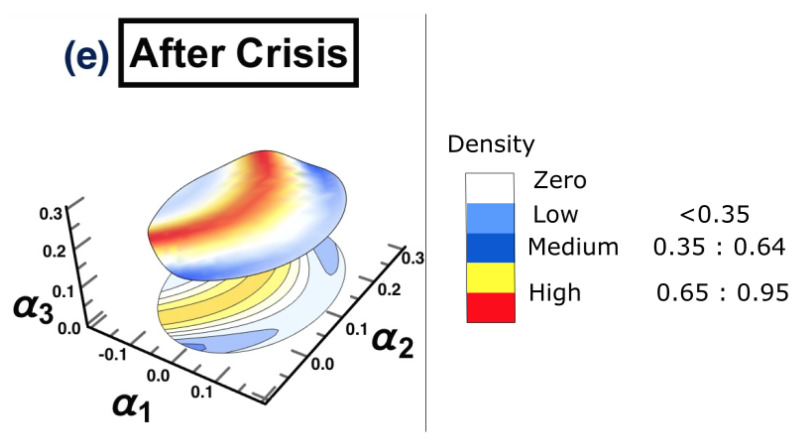
The constructed stock-market three-dimensional interaction parameter space, when a crisis has ended for period (e) that represents the period (2009–2010). The figure shows that there is a region presented as a red curve where the topology of the density structure has changed from the ones observed before the crisis.

**Figure 13 entropy-25-00070-f013:**
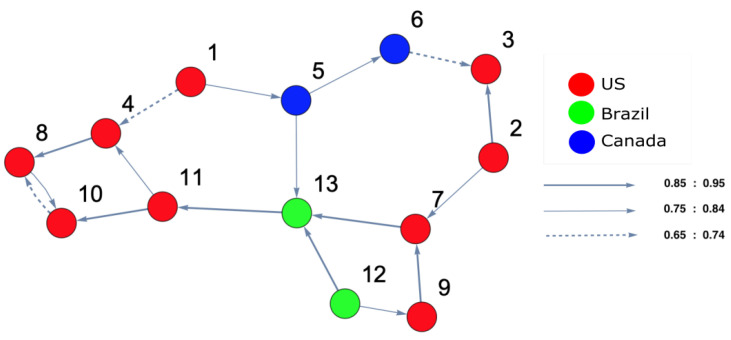
The constructed stock-market interaction network structure. The constructed network reflects the interaction structure among markets, which corresponds to the high-density region, which spans the red region along the directed arrow in the constructed space presented in Figure 12. The red-colored nodes correspond to U.S. stock markets, the blue-colored nodes correspond to stock markets belonging to Canada and the green-colored nodes correspond to stock markets belonging to Brazil. The network shows that there is no interesting pattern to be captured.

**Figure 14 entropy-25-00070-f014:**
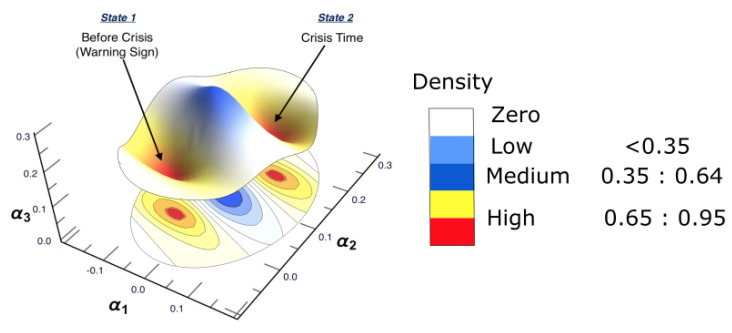
A summary graph of the results. The figure presents the combination of the three-dimensional interaction parameter spaces presented in Figure 6 and Figure 10. It shows that there is a transition occurring between two states, mainly the time before the crisis and the crisis time, forming two holes of clusters. This explains when the interactions among markets reach their peak, which, in turn, can be considered an indication that there is a potential crisis.

**Figure 15 entropy-25-00070-f015:**
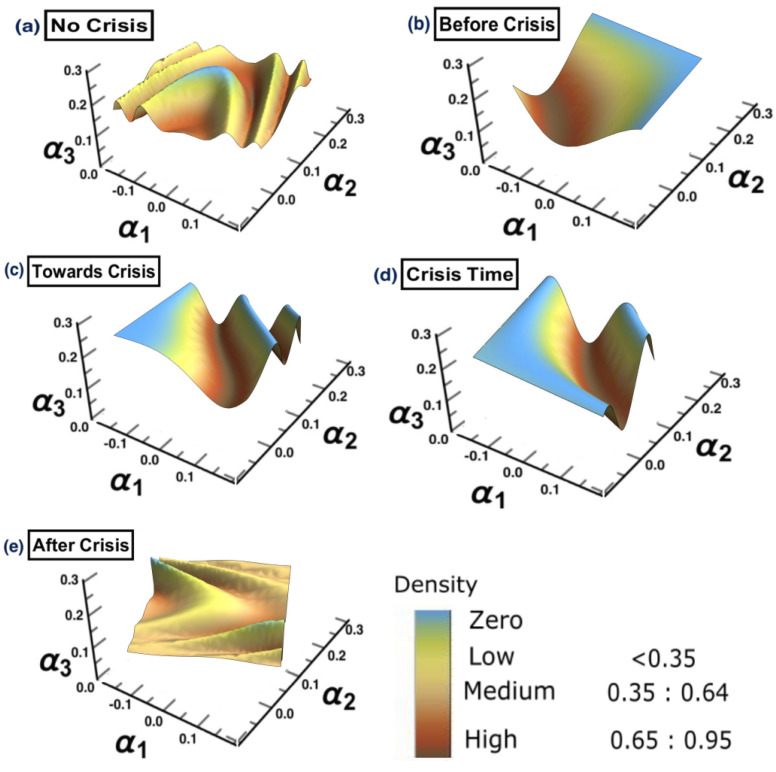
The constructed stock-market three-dimensional interaction parameter space. This is for all stages of the effect of COVID-19 crisis on stock markets. More precisely, sub-figure (**a**) shows the topology structure where no crisis occurs for the time period (1/10/2018 to 31/3/2019), sub-figure (**b**) shows the topology structure for the time period before crisis (1/4/2019 to 31/10/2019), sub-figure (**c**) shows the topology structure for the time towards crisis (1/11/2019 to 28/2/2020), sub-figure (**d**) shows the topology structure where the crisis occurs for the time period (1/3/2020 to 31/12/2020) and the final stage, the sub-figure (**e**) shows the topology structure for the time after crisis (2021). The figure shows how the topology of the interaction surface has changed from one state to another. The density regions, represented in dark brown, present the highest interactions between interaction coefficients 0.65 and 0.95. The transition occurred in stage (**b**) where the markets were towards falling into a crisis forming a clear cluster shown in stage (**d**).

**Table 1 entropy-25-00070-t001:** Results for order selection using AIC${}_{i}$. The table shows that the optimal order for the SSM is three. More precisely, the true model order corresponds to the largest AIC${}_{i}$ value.

Order	AICi
1	49
2	42
3	732
4	34
5	20
6	5
7	0
8	0
9	0
10	0

**Table 2 entropy-25-00070-t002:** Degree centrality. This table presents the result of the calculation of out-degree and in-degree for each node separately corresponding to the node number explanation of Figure 9. The table shows that nodes 1, 5, 10 and 16 are the most influential nodes in the interaction network corresponding to the highest out-degree, while nodes 3 and 8 are the most affected nodes corresponding to the highest in-degree.

Node Number	Out-Degree	In-Degree
1	3	0
2	2	2
3	1	3
4	1	1
5	3	2
6	1	2
7	0	1
8	1	3
9	2	2
10	3	1
11	2	2
12	1	1
13	1	1
14	2	1
15	1	2
16	3	1
17	0	2

**Table 3 entropy-25-00070-t003:** Degree centrality. This table presents the result of the calculation of out-degree and in-degree for each node separately, corresponding to the node number explanation of Figure 11. The table shows that nodes 1, 5, 10 and 16 are the most influential nodes in the interaction network corresponding to the highest out-degree, while nodes 3, 6 and 8 are the most affected nodes corresponding to the highest in-degree.

Node Number	Out-Degree	In-Degree
1	3	0
2	2	2
3	2	3
4	1	1
5	3	2
6	1	3
7	1	1
8	1	4
9	2	2
10	3	1
11	2	2
12	1	1
13	1	1
14	2	1
15	1	2
16	3	1
17	0	2

**Table 4 entropy-25-00070-t004:** Results for order selection using AIC${}_{i}$. The table shows that the optimal order for the SSM is three. More precisely, the true model order corresponds to the largest AIC${}_{i}$ value.

Order	AICi
1	53
2	34
3	945
4	56
5	28
6	3
7	0
8	0
9	0
10	0

## Data Availability

The dataset utilized in this manuscript is available and can be obtained from the Yahoo Finance website [84]. Note that, after clicking on the specified index, historical data should be chosen and the dates under study determined.

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
