# Peer review of "Improving the Process of Early-Warning Detection and Identifying the Most Affected Markets: Evidence from Subprime Mortgage Crisis and COVID-19 Outbreak—Application to American Stock Markets"

_entropy, 2022, doi:10.3390/e25010070_

Round 1
Reviewer 1 Report
The article uses some sophisticated filtering of time series in order to analyze some topology of indices belonging to stock markets of the Americas. The density of the article makes it difficult to follow, mainly in the beginning, but everything seems well explained to me.
What I found the most interesting in the paper is that it clearly shows, in the topology generated by three parameters of the model, a transition from a topology pre crisis of 2008 to a topology during the crisis. It also shows that the topology post crisis is different from the one pre crisis and from a period with no crisis.
Also, it was interesting the way causal networks (in the understanding of Granger causality) were created in order to explain some relations between the indices of the Americas.
Suggestions and corrections
1 – Although the Introduction is usually read after the Abstract, it is not a continuation of it. So, you should not start your introduction with “On the other hand”.
2 – Why is noise assumed to be Gaussian in the SSM model? Is this a simplifying assumption? Is it suitable to stock markets?
3 – Even thought the article describes the causal networks in length, I think that a discussion of the interactions of the US indices would be in order. There are discussions of interactions among indices from different countries, but there is a rich and unexplored structure within the US indices.
English corrections and typos
Please, change, according to your own discretion:
· “existing nave parameter estimators” to “existing naïve parameter estimators” (I am guessing, here).
· “Therefor” to “Therefore”.
· “more deep” to “deeper”.
· “Explicity, This Maximum” to “Explicitly, this Maximum”.
· “between markets of which most strongly affected” to “between markets that are most strongly affected”.
· “matrix Ctogehter” to “matrix C together”.
· “using parameters” to “using as parameters”.
· “the kalman filter” to “the Kalman filter”.
· “belonging to Brazil country” to “belonging to Brazil”. The same for Canada, Panama, and Colombia.
Author Response
Many thanks for your taking the time to review my manuscript, and many thanks for your great comments. I really do appreciate these critical comments, which are a reason for improving the quality of the manuscript. The responses to all comments have been attached point by point.
Note: All the changes done on the manuscript are as follows. The new stuff written in red and other lines have been deleted.

Reviewer 2 Report
The authors submitted a paper related to time series analysis, particularly crises detection in financial time series.
Unfortunately, the paper suffers serious weaknesses.
Starting from the terminology point of view, the analysis is done on the stock markets indexes describing the situation of some sectors, not stock markets.
In general, the paper is poorly written. It begins with the "On the other hand..." expression. Which is nonsense in this place. There are missing references. Strange grammar, abbreviation defined after its usage. Many serious mistakes make the paper difficult in reading.
Despite the language, the method is not properly justified. The authors claim that it is capable of crisis recognition (and prediction!) based only on one example analysis with very arbitrarily chosen time series. Such a hypothesis requires much deeper analysis and justification. It is absolutely not clear that (not defined density) is sensitive to crises.
The necessary literature review is also missing particularly in the area of crisis detection and measurement methods. There are many methods of crisis detection. The authors should mention them and provide its features comparison. Particularly advantages and disadvantages, area of application, weak points etc. Without appropriate reference to known and used methods, the article cannot be accepted for publication.
Author Response
Many thanks for your taking the time to review my manuscript, and many thanks for your great comments. I really do appreciate these critical comments, which are a reason for improving the quality of the manuscript. The responses to all comments have been attached point by point.

Reviewer 3 Report
Referee report on: “Improving the process of early warnings detection and identifying the most affected markets: Application to American stock markets” (entropy-1911686)
This paper proposes the use of the ARE algorithm for the analysis of “early warnings” in equity markets for a sample that spans 14 countries. The authors argue that the method provides a “deep understanding” around the financial crisis of 2008-2009.
Please see my comments below:
1) My first concern is that the paper does not apply any kind of entropy on financial markets. Therefore, it is outside the scope of the journal (Entropy) and should be desk rejected.
2) Then, the paper is unclear and confusing on many points. For instance, on page 8, it mentions “41 American stock markets’ indices of 14 countries”!? How can “American stock market indices” exist in 14 different countries? What are these indices? Why did not you pick the main stock indices of each country for your study? Then, why did you use these particular (14) countries? There has to be some logic behind the sample choice? (e.g., use E.U. vs non-E.U. countries) Otherwise, the analysis and its conclusions do not make sense.
3) The paper claims that one of its contributions is the method’s ability to process non-stationary variables (equity indices), but at the same time it applies the AIC which requires stationary time series. I would prefer to have the analysis done with stationary variables, i.e., stock market returns. This is standard in the literature.
4) Finally, the results are chaotic, to say the least. They are mechanically stated and no economic interpretation is provided. What does it mean to state that “…the US, Brazil, Canada, and Colombia are the most affected markets”? What do we earn from this? Why are these markets the most affected? We already know that the U.S. was the most affected country because the crisis had originated there.
5) The causality diagrams are super confusing and no significant discernible patterns are visible. Are these causality links statistically significant? How did you measure statistical significance in your work?
6) We already know a lot about the subprime mortgage crisis. It would be more beneficial to consider the impacts of the COVID-19 pandemic or the Ukraine crisis on financial markets!
7) The paper must be proofread by a native speaker. It would benefit from a touch of an English speaker who is a financial expert to fix the terminology.
Author Response

(The authors gave the same response as above.)

Reviewer 4 Report
Author of the manuscript “Improving the process of early warnings detection and identifying the most affected markets…” applies the method of autoregressive estimation introduced in Ref. 30 to financial data for the first time. Author claims that this method allows for distinguishing different phases of a financial market evolution: a pre-crisis, crisis, and normal one.
I find this manuscript potentially interesting as an early detection of an oncoming crisis would be of high practical importance for both the investors and the policy-makers. However, for it to be actually interesting and worth reading, Author is asked to address the following two major issues.
-
Conclusions based on the results presented in the manuscript seem to be too general. Only one crisis was analysed, so there is a risk that if some other crisis time had been included, the results would have been different. As the studied data spans the interval 2006-2010, which from today’s perspective seem rather distant, it is natural to expect that Author will extend their analysis to cover also more recent periods. For example, the Covid-19 outburst and the recent market drops offer an excellent possibility to include another market crisis. The prospective outcomes of such an extended analysis covering at least two crises and a long normal market phase could be of more scientific and practical value.
-
The formalism applied in the manuscript is rather sophisticated. Could similar results be obtained by using some simpler approach? Transfer entropy and volatility spillover are examples of the measures that could potentially offer similar functionality. Author is asked to provide a relevant discussion.
Minor remarks are the following:
-
There are symbols in Eq. (1) that differ from their description below. They should be made more consistent.
-
In Eq. (3) the letters X and x denote the same process under study, so why are two symbols used?
-
It is recommended to distinguish scalars and vectors/matrices in Eqs. (1)-(13) by using normal font for the former and bold font for the latter. It will facilitate comprehension of these equations.
-
Figs. 3, 4, 6, 8, 10 & 12 would be much more reader-friendly if the nodes were labelled by an associated index acronym instead of numbers. Then Tables 2-7 could be removed as unnecessary.
-
The colors and color scales have to be agreed in different figures. I do not see why there is no gradual transition from white to red via blue and yellow. Instead, in many cases there is a sequence: blue-white-yellow-red which seems counterintuitive if compared with the associated color scale.
-
The node colors have to be more distinguishable on a B&W hardcopy, now they are not. Author should take care of readers who read hardcopies.
-
There are some typos, for example, lines 43 (naive), 44 (therefore), 95 (Kalman filter), 106 (C together), 115 (Akaike), 148 (Kalman) that need correction.
-
Referencing in the text body is sometimes incorrectly structured, e.g., in lines 46, 115+, 118+, 182+ (Eichler).
-
Ref. 29 lacks journal information.
-
There are instances of using incorrect forms like “Brazil country”, “Canada country”, etc.
-
The word “thesis” is used in line 182+, which might not be relevant in a journal publication.
The manuscript can be considered for publication only after Author addresses the major issues and correctz it according to the list of minor issues.
Author Response

(The authors gave the same response as above.)

Round 2
Reviewer 1 Report
The article has been rewritten, taking into account all the reviewers' suggestions. Particularly, the discussion of result has been greatly improved.
Author Response
I really do appreciate all your effort, spending time to read my work. All your comments have greatly improved the quality of the manuscript very well. Thanks a lot.

Reviewer 2 Report
The authors significantly improved the quality of the presentation.
However, in the introduction, there are two serious mistakes.
1. l. 83 Log of the returns does not affect stationarity. It is used to remove spurious trends related to the random walk.
2. l.93 not "long tails" but have tails in the case of short returns. The long-time returns follow Gaussian distributions.
Assuming that the mentioned issues can be easily corrected by the authors the paper can be accepted for publication.
Author Response

(The authors gave the same response as above.)

Reviewer 3 Report
Thank you. The paper has been significantly improved. Also, I understand that the Editor has no concerns about the paper's fit in Entropy, so I will not insist on the issue.
One last addition to the paper would be to cite and briefly discuss in a paragraph research contributions that extract early warning signals of market distress from option contracts:
Bates, D. S. (1991). The crash of ’87: Was it expected? The evidence from options markets. Journal of Finance, 46, 1009–1044.
Bates, D.S. (2000). Post-'87 crash fears in the S&P 500 futures option market,
Journal of Econometrics, Volume 94, Issues 1–2, 181-238.
Gençay, R., & Gradojevic, N. (2017). The tale of two financial crises: An entropic perspective. Entropy, 19(6), 244.
Gradojevic N. (2021). Brexit and foreign exchange market expectations: Could it have been predicted? Annals of Operations Research 297, 167–189.
Xing, Y., Zhang, X., & Zhao, R. (2010). What does the individual option volatility smirk tell us about future equity returns? Journal of Financial and Quantitative Analysis, 45(3), 641–662.
Author Response
I really do appreciate all your effort spending time to read my work. All your comments have greatly improved the quality of the manuscript very well. Thanks a lot.
Reviewer 4 Report
In its modified version, the manuscript still lacks enough scientific soundness to be published as it is. Author hasn't yet included a parallel analysis of any other crisis like the Covid-19 outbreak or so and keeps claiming that his method is robust enough to identify other crises as well. In my opinion the Author's statement that his method allows for an early-stage detection of a financial crisis is too strong to be based solely on a single example. Thus, I insist that the Author provides also other examples to assess with greater certainty the robustness of his methodology. The length of the paper is not a problem at all.
Another issue is that Author hasn't introduced any changes to the equations, in which the symbols x and X seem to be used interchangeably despite his response that he did so.
Finally, the colour scales of Figs. 3, 6, 8, and 10 are still different than what is expected from the figures. If the smallest values are denotes by white colour, the medium ones by blue, and the highest ones by orange/red, the same sequence is expected to be seen in the plots (i.e., white->blue->orange->red), which is not the case (the actual sequences are: blue->white->orange->red). Author is asked to explain these counterintuitive sequences.
Author Response

(The authors gave the same response as above.)
